# Integrated Assessment Method of Emergency Plan for Sudden Water Pollution Accidents Based on Improved TOPSIS, Shannon Entropy and a Coordinated Development Degree Model

**Yan Long [1,\*], Yilin Yang [2], Xiaohui Lei [1], Yu Tian [1] and Youming Li [3]**

[1] China Institute of Water Resources and Hydropower Research, Beijing 100038, China; lxh@iwhr.com (X.L.); sweetrain511@163.com (Y.T.)

[2] State Key Laboratory of Hydraulic Engineering Simulation and Safety, Tianjin University, Tianjin 300350, China; yangyilin0716@tju.edu.cn

[3] BGI Engineering Consultants LTD, Beijing 100038, China; 15320176579@163.com

\* Correspondence: hebeilongyan@163.com

**Abstract:** Water is the source of all things, so it can be said that without the sustainable development of water resources, there can be no sustainable development of human beings. In recent years, sudden water pollution accidents have occurred frequently. Emergency response plan optimization is the key to handling accidents. Nevertheless, the non-linear relationship between various indicators and emergency plans has greatly prevented researchers from making reasonable assessments. Thus, an integrated assessment method is proposed by incorporating an improved technique for order preference by similarity to ideal solution, Shannon entropy and a Coordinated development degree model to evaluate emergency plans. The Shannon entropy method was used to analyze different types of index values. TOPSIS is used to calculate the relative closeness to the ideal solution. The coordinated development degree model is applied to express the relationship between the relative closeness and inhomogeneity of the emergency plan. This method is tested in the decision support system of the Middle Route Construction and Administration Bureau, China. By considering the different nature of the indicators, the integrated assessment method is eventually proven as a highly realistic method for assessing emergency plans. The advantages of this method are more prominent when there are more indicators of the evaluation object and the nature of each indicator is quite different. In summary, this integrated assessment method can provide a targeted reference or guidance for emergency control decision makers.

**Keywords:** coordinated development degree model; technique for order preference by similarity to ideal solution; Shannon entropy; sudden water pollution; integrated assessment; inhomogeneity

## 1. Introduction

Sustainability has become an important topic in the political agenda of many countries [1–3]. Water plays an extremely important role in human survival and regional socioeconomic development, particularly in arid and semi-arid areas [4]. How to use the water resources in a sustainable way is an important problem for sustainable development, especially in areas where water resources are extremely scarce. [5–8]. One of the important factors that restricts the sustainable development of water resources is uneven spatial temporal distribution [9]. Therefore, a continuously growing number of water transfer projects have been built in recent years [10], such as the Central Arizona Project in the United States [11] and the South-North Water Transfer Project (SNWTP) in China [12]. Open channels,

complex operation condition, higher water quality requirement, diverse types and an increasing number of water pollution risk sources are the characteristics of these water transfer projects (WTPs). However, these buildings have greatly increased the pollution risk of water transfer projects [13].

Unlike other pollution accidents, sudden water pollution accidents have varied and complex pollution sources [14,15], while the pollution pathway and degree are unpredictable [16]. Also, sudden water pollution accidents not only cause significant economic losses, but also seriously endanger the safety of industrial and domestic water supplies, thus causing social panic [17,18]. Therefore, there is much research on how to deal with sudden water pollution accidents in WTPs. Lei et al. (2018) [19] developed and implemented a decision support system that can provide technical support in managing pollution accidents. Xu et al. (2016) [20] proposed a real-time, rapid emergency control mode. Wang et al. (2018) [21] proposed a method for identifying pollution sources in rivers that could quickly control the spread of pollution. Long et al. (2016) [22] proposed a more comprehensive emergency control system. Tang et al. (2015) [23] proposed a set of reasonable risk forecasting models based on an integrated Bayesian Network and water quality model. These systems offer valuable warning and response tools to reduce the impact of accidents. But for accidents that have already occurred, these models and systems are less effective. At this time, a more comprehensive plan library is needed, for which the key step is to select the optimal plan quickly and accurately from a plan library.

In the field of evaluation, decision-making models based on statistical data of historical events and subjective expert judgment have been widely used. In the decision-making model, the rationality of the emergency plan for sudden water pollution accidents is determined by analyzing the important indicators affecting sudden water pollution accidents through the technique for order preference by similarity to ideal solution (TOPSIS) [24–26], fuzzy fault tree analysis [27,28], the vulnerability model [29], multi-criteria analysis [30], and other decision methods. These methods are used in different situations. Compared with other multi-criteria decision analysis methods, TOPSIS has some advantages, such as the most adequate use of raw data information and no strict restrictions on the number of indicators [31]. Due to its practicability and scientific nature, TOPSIS has been verified and applied by experts in many fields [32–34]. Unfortunately, only a handful of experts have applied TOPSIS to assess emergency plans and generate decision support options. Therefore, we tried to apply TOPSIS to calculate the relative closeness of the ideal solution.

According to the wooden bucket theory [35,36], the merits of an organization depend not only on the overall conditions, but also on every element of the organization. Just like the two companies participating in an election, the overall score can be the same, but the first company is more average in all aspects, while another company has obvious disadvantages. In this case, the first company will win. Therefore, the unevenness between the indicators of the evaluation object will influence emergency plan assessment and warrants consideration. Accordingly, this research utilizes the Shannon entropy method (1948) to evaluate the inhomogeneity among the attributes of one evaluation object [37].

For the relationship between multiple systems, the coordinated development degree model (CDDM) has been investigated [22,38]. In this paper, relative closeness and the inhomogeneity are regarded as two systems that influence the optimality of the emergency plan and the CDDM is used to assess their relationship.

The emergency plan assessments for global water transfer projects as well for the projects of China have been gradually increasing in quantity. Nevertheless, the existing emergency plan evaluation methods only focus on the final scores of different evaluation objects, and do not reflect the different quality between these evaluation objects. Therefore, this study develops an integrated assessment method of emergency plans for sudden water pollution accidents based on an improved TOPSIS and Shannon entropy method, which can effectively compensate for this drawback.

The method creates a combination of multi-indices system and the inhomogeneity of multiple indicators to assess the emergency plan. Our work aims to (1) establish a multi-indices system of risk, timelines, economy and feasibility; (2) determine the weight of each indicator based on AHP and entropy method; (3) develop an integrated emergency plan assessment method based on

improved TOPSIS and Shannon entropy; (4) verify the rationality of the method with a case study; (5) compare the integrated method with other methods; (6) discuss the advantages and deficiencies of the proposed method.

## 2. Integrated Assessment Method of Emergency Plan

Our study proposes a multiple emergency plan assessment method, which divided into five phases. Firstly, the evaluation indicator system for emergency plan is established by literature research and statistics. Secondly, the weight of each indicator is determined based on AHP and the entropy method. Thirdly, the relative closeness to the ideal solution is calculated based on TOPSIS. Then, the inhomogeneity of the different indicators is assessed based on Shannon entropy. Finally, the emergency plan is comprehensively evaluated using the coordinated development degree model. The framework of the emergency plan assessment method is shown in Figure 1.

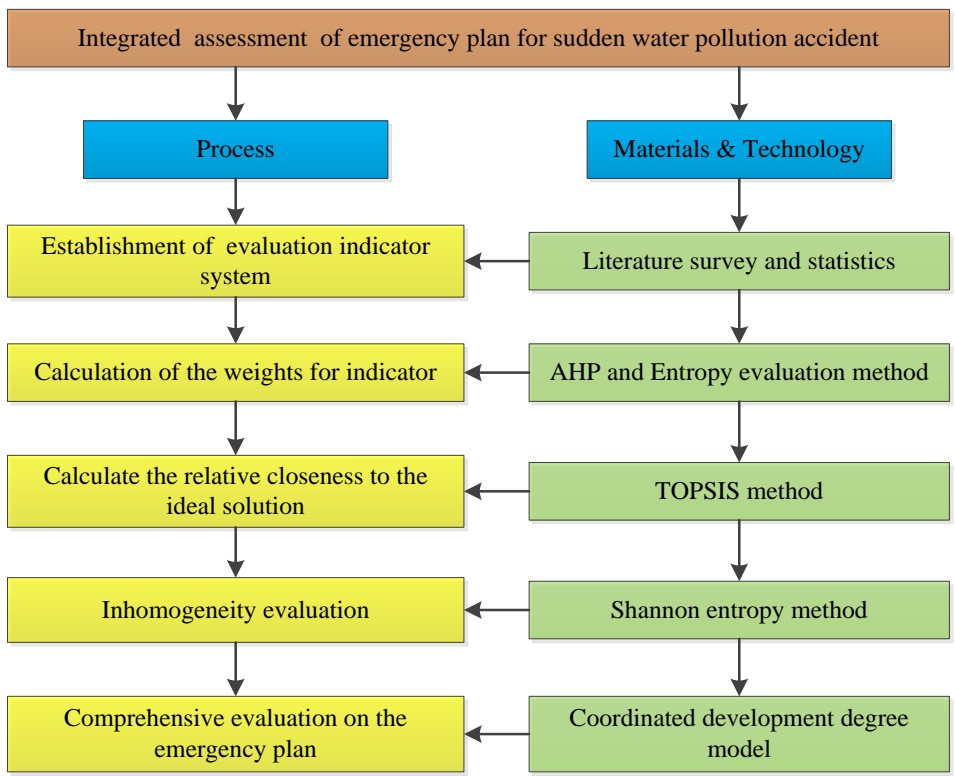

**Figure 1.** The framework of the emergency plan assessment method.

### 2.1. Establishment of Indicator System

Based on the literature review for emergency plan indictors [39,40] and the concept of decision making, which is one of the most important part of modern decision science, the indicator system for emergency plan assessment, is established. The hierarchy structure of the emergency plan assessment is shown in Figure 2, which includes indicators in resistance risk ($B_1$), timeliness ($B_2$), economy ($B_3$) and feasibility ($B_4$). Resistance risk mainly refers to the ability to respond to emergencies, and resistance risk indices here are focused on comprehensiveness of prevention measures ($C_{11}$), accuracy of potential risk assessment ($C_{12}$) and the operator's emergency response to danger ($C_{13}$). Timeliness mainly refers to the speed of the response to an emergency, and here are focused on the speed of the emergency starting ($C_{21}$), speed of arrival of personnel ($C_{22}$) and supplies and speed of restoring water delivery ($C_{23}$). Economy refers to the implementation effect and resource consumption. The main concerns are emergency funds ($C_{31}$), utilization of emergency resources ($C_{32}$) and implementation effects ($C_{33}$). Feasibility mainly refers to the difficulty degree of implementing emergency plans, and

this paper mainly considers the rationality and scientific aspects of the plan ($C_{41}$), controllability of resource allocation ($C_{42}$), contingency response to accident evolution ($C_{43}$) and coordination between departments ($C_{44}$). The factors affecting each indicator are shown in Table 1. When assessing the risk of pollution accidents, we need to consider the types of pollutants, origin of pollution, levels of pollutants, extent of the pollution, location of accidents, economic losses, environmental impacts, and engineering damage.

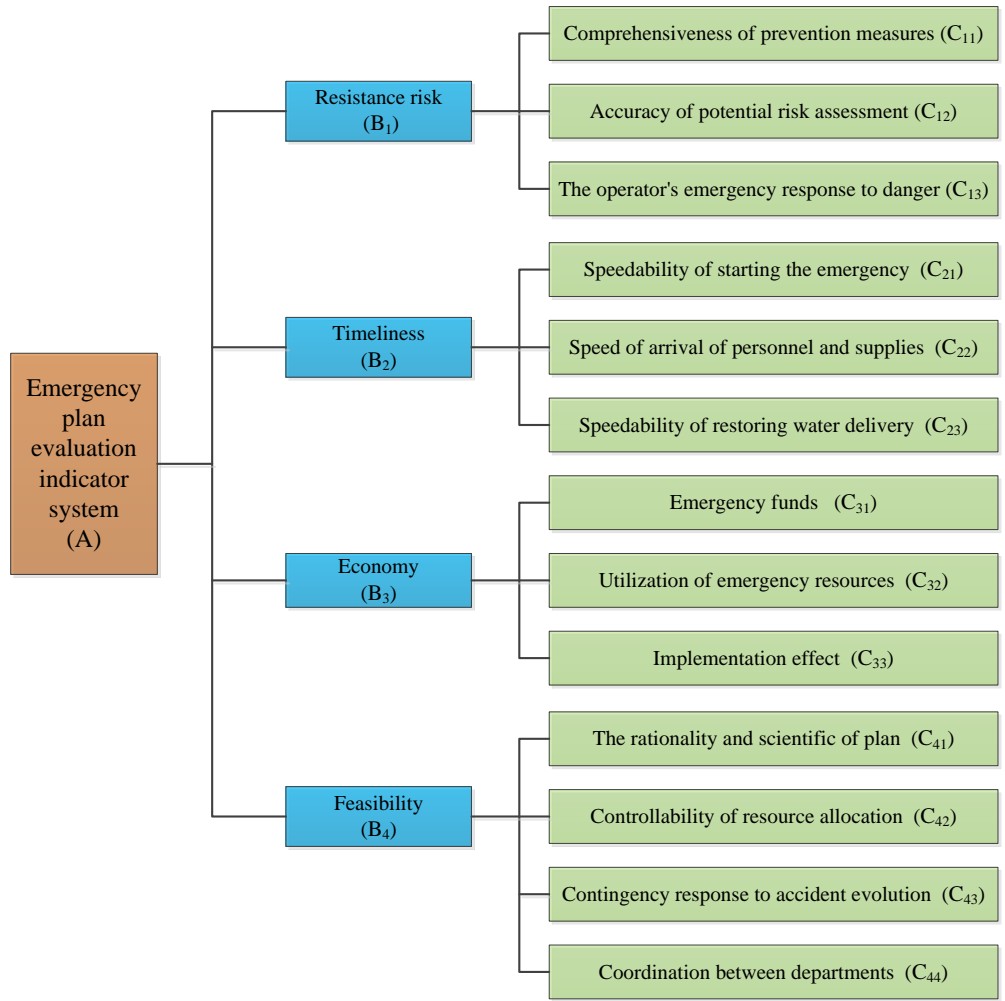

**Figure 2.** Emergency plan evaluation indicator system.

**Table 1.** Description of indicator.

| Indicator | Influence Factor |
|-----------|------------------|
| $C_{11}$ | Flexibility of the plan, Alternative measures, Variety of risks |
| $C_{12}$ | Risk identification accuracy, Risk level accuracy |
| $C_{13}$ | Number of people, Personnel quality, Personnel experience |
| $C_{21}$ | Program complexity, Ease of operation |
| $C_{22}$ | Traffic, Distance, Arriving time |
| $C_{23}$ | Time of stabilization, Number of check gates which need to adjust, Maximum operating time of a single check gate, Total operating time of all the check gates |
| $C_{31}$ | Labor fee, Equipment cost, Material costs, Loss cost |
| $C_{32}$ | Material utilization rate, Staff utilization, Equipment usage rate |
| $C_{33}$ | Water quality level after disposal, The number of abandoned waters, Damage degree of water transfer project |
| $C_{41}$ | Technical maturity, Operability, Rationality, Scientific basis, Degree of acceptance |
| $C_{42}$ | Number of materials and personnel, Allocation rights |
| $C_{43}$ | Variability, Resilience |
| $C_{44}$ | Co-movement, Coordination, Communication ability |

*2.2. Calculation of the Weights for the Indicator*

At present, there are many methods to determine the indicator weight, which can be roughly divided into subjective weighting assessment, objective weighting assessment and combined integration weighting assessment. The subjective weighting method refers to the method of comparing, assigning and calculating the weight of each indicator according to the knowledge and experience or preferences of the decision makers, among which AHP [41,42] and Delphi [43,44] are commonly chosen. The objective weighting method is a method for determining the weight of each indicator based on the difference in objective data of each program evaluation indicator value. It mainly includes principal component analysis [45,46], the mean squared error method [47,48], and the entropy method [49,50], and so on. No matter which method is selected, the weight obtained will have a certain deviation, so in order to compensate for this defect, two weight methods need to be combined, which is the combined integration weighting method. To this end, the AHP and entropy weight method are combined to determine the indicator weights [25]. The main steps of the combined integration weighting method are summarized as follows:

(1) Constructing a raw data matrix of *m* objects and *n* indices is $R = (x_{ij})_{m \times n}$

$$R = \begin{bmatrix} x_{11} & x_{12} & \cdots & x_{1n} \\ x_{21} & x_{22} & \cdots & x_{2n} \\ \cdots & \cdots & \cdots & \cdots \\ x_{m1} & x_{m2} & \cdots & x_{mn} \end{bmatrix} \tag{1}$$

(2) Calculating the weight $p_{ij}$ of the *j*th indicator:

$$p_{ij} = y_{ij} / \sum_{i=1}^{m} y_{ij} \tag{2}$$

(3) The entropy $e_j$ of the *j*th indicator is defined as follows:

$$e_j = -\frac{1}{\ln m} \sum_{i=1}^{m} p_{ij} \ln p_{ij} \tag{3}$$

(4) Calculating the deviation degree $g_j$ of the *j*th indicator:

$$g_j = 1 - e_j \tag{4}$$

(5) The indicator of entropy weight is defined as follows:

$$\mu_j = \frac{g_j}{\sum_{j=1}^{n} g_j} \tag{5}$$

(6) We calculate the subjective weight $\lambda_j$ by using the AHP method. The detailed steps of AHP method are listed in Long and Xu (2016) [22];

(7) We calculate the final weight $\omega_j$ as follows:

In order to make the distribution of objective weight and subjective weight more reasonable, the distance function is introduced to combine the subjective weight with the objective weight. The distance function between the subjective weight $\lambda_j$ and the objective weight $\mu_j$ is:

$$D(\lambda_j, \mu_j) = \left[ \frac{1}{2} \sum_{j=1}^{n} (\lambda_j - \mu_j)^2 \right]^{0.5} \tag{6}$$

The subjective weight and the objective weight distribution coefficient are $\alpha$ and $\beta$, respectively, and the calculation is performed using Equations (7) and (8).

$$D\left(\lambda_j, \mu_j\right)^2 = (\alpha - \beta)^2 \tag{7}$$

$$\alpha + \beta = 1 \tag{8}$$

The combined integration weighting $\omega_j$ of the $j$th indicator is calculated as follows:

$$\omega_j = \alpha\lambda_j + \beta\mu_j \tag{9}$$

*2.3. Calculate the Relative Closeness to the Ideal Solution Based on TOPSIS*

Compared with other multi-criteria decision analysis methods, TOPSIS has some advantages, such as the most adequate use of raw data information and no strict restrictions on the number of indicators [31]. TOPSIS is a ranking method that attempts to choose alternatives that simultaneously have the shortest distance from the positive ideal solution and the farthest distance from the negative ideal solution [51]. TOPSIS makes full use of the attribute information and provides a cardinal ranking of the alternatives [52]. The TOPSIS method was first proposed by C. L. Hwang and K. Yoon in 1981. The TOPSIS method is based on the closeness of a limited number of evaluation objects to the ideal solution [52]. There are two ideal solutions, one is the positive ideal solution, and the other is the negative ideal solution. The best object should be the closest to the positive ideal solution and the farthest from the negative ideal solution [51]. In this study, the relative closeness to the ideal solution is calculated based on TOPSIS.

The calculation process of the TOPSIS method is as follows [34]:

Step 1: Constructing the original decision matrix:

In Figure 2, the determination of each indicator is related to a number of factors, and the score $C_i$ of each indicator is determined in the form of expert consultation, as shown in Equation (10).

$$C_i = 0.5 \times \left[1 + \sum_{k=1}^{q} S_k / (q \times 10)\right] \tag{10}$$

Where $S_k$ is the expert score of the $k_{\text{th}}$ factor which affect the $i$th indicator.

According to the calculated value, the number of evaluation objects is denoted by $m$. At the same time, the number of evaluation indicators is recorded as $n$. The performance matrix can be presented as Equation (1).

Step 2: Normalize the decision matrix:

Because the evaluation indicator value types are different, the values can be accurate real values, interval numbers, triangular fuzzy numbers, etc. Therefore, the evaluation indicators in this paper are uniformly normalized by the following Equation.

$$r_{ij} = x_{ij} / \sqrt{\sum_{i=1}^{m} x_{ij}^2} \tag{11}$$

Step 3: Calculate the weighted normalized decision matrix:

$$c_{ij} = \omega_j r_{ij} \tag{12}$$

where $\omega_j$ is the combined integration weighting of the $j$th indicator.

Step 4: Calculate the distance from each alternative to the positive ideal solution and the negative ideal solution:

$$d_i^+ = \sqrt{\sum_{j=1}^{n} (c_{ij} - c_j^+)^2}, i = 1, 2, \dots, m \tag{13}$$

$$d_i^- = \sqrt{\sum_{j=1}^{n} (c_{ij} - c_j^-)^2}, i = 1, 2, \dots, m \tag{14}$$

where $d_i^+$ denotes the distance between the *i*th alternative and the positive ideal solution, while $d_i^-$ denotes the distance between the *i*th alternative and the negative ideal solution, where $c_j^+ = \max$ ($c_{ij}, i = 1,2 \dots m$) and $c_j^- = \min (c_{ij}, i = 1, 2 \dots m)$.

Step 5: Calculate the relative closeness to the ideal solution:

$$V_i = d_i^- / (d_i^- + d_i^+), i = 1, 2, \dots, m \tag{15}$$

*2.4. Inhomogeneity Evaluation Based on the Shannon Entropy Method*

The optimal solution determined by TOPSIS is that it does not consider the inhomogeneity of evaluation indicator. The inhomogeneity of the evaluation indicators and the non-hierarchical value of the indicator weights will affect the determination of the optimal emergency plan.

In this study, the Shannon entropy method is employed to calculate the inhomogeneity of all indicators. In 1948, Shannon proposed the concept of "information entropy" to solve the problem of quantitative measurement of information. The amount of information in a piece of information is directly related to its uncertainty. For example, if we want to figure out a very uncertain thing, or something we don't know, we need to know a lot of information. On the contrary, if we have a lot of knowledge about something, we don't need much information to figure it out. So, from this perspective, we think the measure of the amount of information is equal to the amount of uncertainty. As proposed by Shannon, Shannon entropy is widely accepted [53,54]. According to Shannon entropy, the larger the discrete degree of an indicator is, the larger influence it may have in the decision-making process, and thus, the smaller the information entropy, the reliability of the scheme, and vice versa. The calculation process is presented as follows:

$$H_i = -\frac{1}{\ln n} \sum_{j=1}^{n} p_{ij} \ln p_{ij}, (i = 1, \dots, m) \tag{16}$$

$$p_{ij} = c_{ij} / \sum_{j=1}^{n} c_{ij} \tag{17}$$

where $H_i$ is the entropy of the evaluation object, *m* is the number of evaluation objects, *n* is the number of attributes indicators.

*2.5. Comprehensive Evaluation Based on Coordinated Development Degree Model*

Based on the above relative closeness and inhomogeneity evaluation results, we try to apply a coordinated development degree model (CDDM) for emergency plan assessment, focusing on the relationship between relative closeness and inhomogeneity of emergency plans. The calculation process is presented as follows:

Step 1: Data normalization to (0,1)

$$V_i' = V_i / V_{\max} \quad H_i' = H_i / H_{\max} \tag{18}$$

Step 2: Coordinated development degree calculation

In order to quantify the relationship between relative closeness and inhomogeneity, the coordinated development degree is used to characterize it.

$$M_i = \left( (V'_i \times H'_i) / \left[ \frac{V'_i + H_i'}{2} \right]^2 \right)^2 \tag{19}$$

$$D_i = \sqrt{M_i T} \tag{20}$$

where $M_i$ is the coordination degree of the *i*th emergency plan; $T$ is the comprehensive evaluation indicator of the system, here $T = 13$; $D$ is the coordinated development degree of the *i*th emergency plan.

Step 3: Determining the optimal emergency plan

According to the above method, the $D$ of each emergency plan is determined, and then the plan is sorted. The larger $D$ is, the better the corresponding emergency plan is, that is:

$$Optimal\ emergency\ plan = \max\{D_i\}, i = 1, 2, \ldots, m \tag{21}$$

## 3. Method Application and Discussions

### 3.1. Scenario Design

The SNWT Project, one of the most important and strategic projects in China, will be beneficial to suppress and reduce groundwater withdrawal. The decision support system for the Middle Route has been deployed by the Middle Route Construction and Administration Bureau since 2010 [19]. Emergency response plans for 388 pollutants have been stored in this system. In order to test the rationality of the method, we extracted a scenario in the system for testing. A schematic diagram of the test area is shown in Figure 3. The detailed descriptions of the designed scenarios are shown in Table 2. According to the rapid prediction formulas proposed by Xu [20], the time for the pollutants to arrive at the Qi River Gate and the Shier-li River Gate was 20 and 180 min respectively. Given that the designed water conveyance flow of the Taocha Dam is 350 m$^3$/s, four emergency plans were proposed, as shown in Table 3.

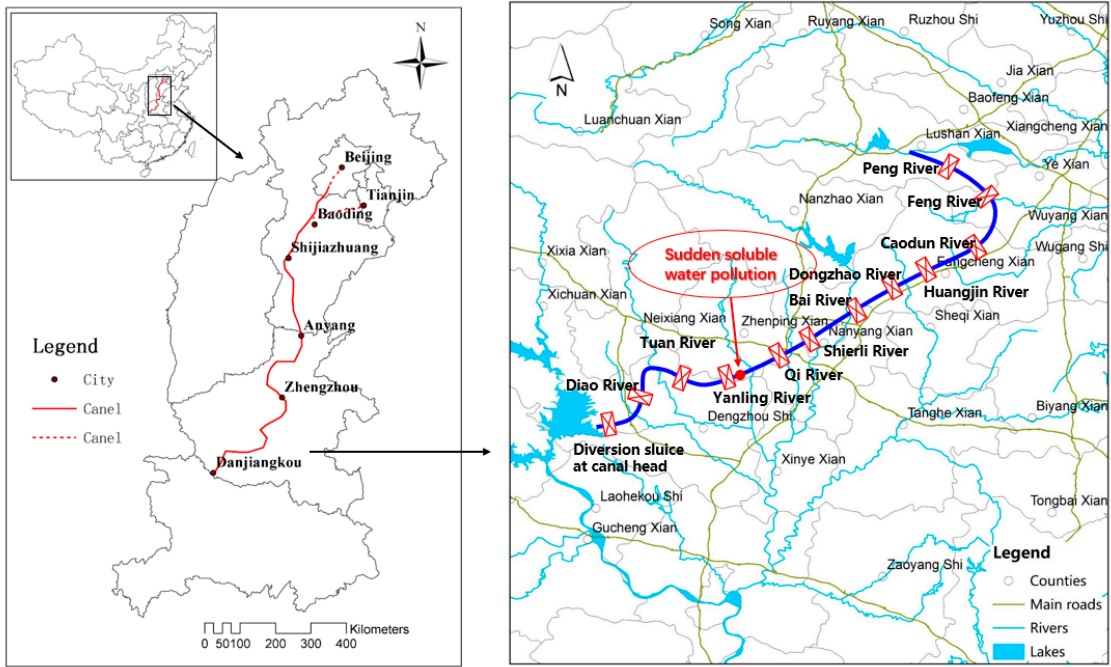

**Figure 3.** Schematic of the test area.

**Table 2.** Scenario set for sudden water pollution accident in the study area.

| Upstream Check Gate | Downstream Check Gate | Pollution Mode | Location of Pollution | Type of Pollutant | Pollutant Weight |
|---|---|---|---|---|---|
| Yanling River | Qi River and Shier-li River | Instantaneous pollution | Central | Non-degradable soluble pollutants | 10 ton |

**Table 3.** Emergency plans.

| Emergency Plan | Emergency Plan | | |
|---|---|---|---|
| | Accident Pool | Upstorm Pool | Downstorm Pool |
| EP 1 | The Yanling River Gate and the Qihe River Gate were closed for 15 min. The Tanzhai water diversion is not activated and no emergency dispatch is required. | The water supply is normally supplied to each water inlet. Each check gate is closed synchronously for 15 min, and each backwater gate is not activated. | Downstream gates maintain the normal water level before the gate. |
| EP 2 | | The water supply is normally supplied to each water inlet. The water level fluctuation before the sluice gates is 30 cm. | |
| EP 3 | The pollutants are confined to the following two channels. The order of closing the gates is: First, the Yanling gate was closed in 15 min. Secondly, after 30 min, the Qi River gate was closed and the closing time was 60 min. Finally, after 120 min, the Shier-li River gate was closed and the gate was closed for 60 min. At the same time, the Weihe River sluice gate is activated. | The water supply is normally supplied to each water inlet. Each check gate is closed synchronously for 15 min, and each backwater gate is not activated. | |
| EP 4 | | The water supply is normally supplied to each water inlet. The water level fluctuation before the sluice gates is 30 cm. | |

Note: EP stands for emergency plan.

## *3.2. Integrated Assessment Method of Emergency Plan*

According to the proposed integrated assessment method of emergency plan, the four kinds of emergency plans discussed previously are evaluated, and the optimal plan is selected. The calculation process and algorithm are shown in Figure 4 and the methodology steps together with the results are explained as follows.

Step 1: Determine the indicator value.

Based on Figure 1 and the principles of Table 1 and on experts' experience and knowledge, the expert score sheets for the four emergency plans are as shown in Tables S1–S4 (Electronic Supplementary Material). According to Equation (10), the data in Tables S1–S4 are calculated to obtain indicator values, as shown in Table S5 (Electronic Supplementary Material). The average value of the five expert scores was taken as the final value of the indicator, and the results are shown in Table 4.

Step 2: Calculation of objective weight $\mu_j$ for the indicator

The data in Table 4 was normalized and then the normalized matrix $Y$ are obtained as shown in Equation (S1) (Electronic Supplementary Material). The entropy of each indicator is obtained according to Equations (2) to (3). The entropy $e_j$ and entropy weight of the indicators $\mu_j$ are obtained according to Equations (4) to (5). The results are shown in Table 5.

Step 3: Calculation of subjective weight $\lambda_j$ for the indicator

We calculate the subjective weight $\lambda_j$ by using the AHP method for taking the subjective attributes of the data into consideration. After expert consultation, the judgment matrices of four kinds of emergency plans are obtained. Take the emergency plan one as an example for detailed introduction. Based on the principles of Table 1 and on experts' experience and knowledge, the calculations of

the local weights of events $B_1$–$B_4$, resistance risk event $C_{11}$–$C_{13}$, timeliness event $C_{21}$–$C_{23}$, economy events $C_{31}$–$C_{33}$, and feasibility events $C_{41}$–$C_{44}$ were shown as the values in Tables S6–S10 (in Electronic Supplementary Material). The local weight and global weight of each evaluation indicator could be achieved according to Equations (1) to (4) in Long's paper (2016).

Step 4: Calculation of integration weighting $\omega_j$

**Table 4.** Indicator value.

| Indicator | Emergency Plan 1 | Emergency Plan 2 | Emergency Plan 3 | Emergency Plan 4 |
|---|---|---|---|---|
| C11 | 0.7994 | 0.7994 | 0.8466 | 0.8466 |
| C12 | 0.91 | 0.91 | 0.91 | 0.91 |
| C13 | 0.8834 | 0.8834 | 0.8834 | 0.8834 |
| C21 | 0.91 | 0.89 | 0.91 | 0.91 |
| C22 | 0.8432 | 0.8432 | 0.8432 | 0.8432 |
| C23 | 0.8125 | 0.795 | 0.78 | 0.705 |
| C31 | 0.805 | 0.8275 | 0.805 | 0.805 |
| C32 | 0.85 | 0.86 | 0.85 | 0.85 |
| C33 | 0.7968 | 0.8564 | 0.8334 | 0.8568 |
| C41 | 0.78 | 0.78 | 0.798 | 0.798 |
| C42 | 0.885 | 0.885 | 0.885 | 0.885 |
| C43 | 0.665 | 0.665 | 0.815 | 0.815 |
| C44 | 0.91 | 0.91 | 0.91 | 0.91 |

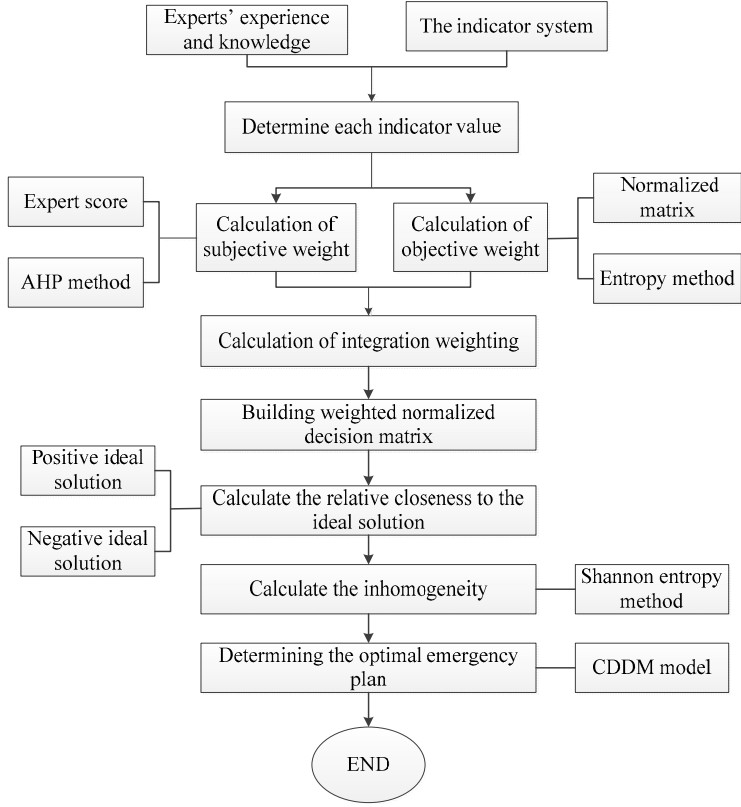

**Figure 4.** Calculation process and algorithm.

**Table 5.** The entropy of each indicator.

| Indicator | $e_j$ | $g_j$ | $\mu_j$ |
|---|---|---|---|
| C11 | 0.522053 | 0.477947 | 0.079665987 |
| C12 | 0.723985 | 0.276015 | 0.046007249 |
| C13 | 0.666842 | 0.333158 | 0.055532036 |
| C21 | 0.715917 | 0.284083 | 0.047352039 |
| C22 | 0.562625 | 0.437375 | 0.072903264 |
| C23 | 0.248989 | 0.751011 | 0.125181358 |
| C31 | 0.446586 | 0.553414 | 0.092245185 |
| C32 | 0.586764 | 0.413236 | 0.068879704 |
| C33 | 0.538518 | 0.461482 | 0.076921594 |
| C41 | 0.38384 | 0.61616 | 0.102703973 |
| C42 | 0.670515 | 0.329485 | 0.054919736 |
| C43 | 0.209997 | 0.790003 | 0.131680655 |
| C44 | 0.723985 | 0.276015 | 0.046007249 |

The integration weighting $\omega_j$ of each indicator is calculated according to Equations (6) to (9). The results are shown in Table 6.

**Table 6.** The integration weighting $\omega_j$ of each indicator.

| Indicator | $C_{11}$ | $C_{12}$ | $C_{13}$ | $C_{21}$ | $C_{22}$ | $C_{23}$ | $C_{31}$ | $C_{32}$ | $C_{33}$ | $C_{41}$ | $C_{42}$ | $C_{43}$ | $C_{44}$ |
|---|---|---|---|---|---|---|---|---|---|---|---|---|---|
| $\omega_j$ | 0.035 | 0.04 | 0.031 | 0.112 | 0.08 | 0.064 | 0.124 | 0.122 | 0.186 | 0.082 | 0.031 | 0.071 | 0.022 |

Step 5: Building a weighted normalized decision matrix

According to Equations (11) to (12), a weighted normalized decision matrix of the indicator is constructed as shown in Equation (S2) (in Electronic Supplementary Material).

Step 6: Calculate the relative closeness to the ideal solution

According to Equations (13) to (14), the distance from each alternative to the positive ideal solution and the negative ideal solution are calculated as shown in Equations (S3) and (S4) (in Electronic Supplementary Material). Then, the relative closeness to the ideal solution of each emergency plan are calculated based on Equation (15), the result is shown in Equation (22).

$$V = \left\{ \begin{array}{cccc} 0.4305 & 0.3507 & 0.3756 & 0.3551 \end{array} \right\} \tag{22}$$

Step 7: Calculate the inhomogeneity

The larger the discrete degree of an indicator is, the larger influence it may have in the decision-making process, and thus, the smaller the information entropy, the reliability of the scheme, and vice versa. The Shannon entropy of each emergency plan is calculated according to Equations (16) to (17), and the result is as shown in Equation (23).

$$H = \left\{ \begin{array}{cccc} 3.209 & 3.134 & 3.219 & 3.172 \end{array} \right\} \tag{23}$$

Step 8: Determining the optimal emergency plan

According to Equation (18), data of relative closeness and Shannon entropy are normalized as shown in Equations (S5) and (S6) (in Electronic Supplementary Material). And then the coordination degree and coordinated development degree are calculated as shown in Equation (S7) (in Electronic Supplementary Material) and Equation (24).

$$D = \left\{ \begin{array}{cccc} 3.5896 & 3.5771 & 3.6055 & 3.5772 \end{array} \right\} \tag{24}$$

The larger $D$ is, the better the corresponding emergency plan is. Therefore, according to Equation (21), the optimal emergency plan is the emergency plan 3.

### 3.3. Discussion

This work presents an approach for assessing the emergency plan by combining improved TOPSIS, Shannon entropy and Coordinated development degree model. The Shannon entropy method was used to analyze different types of index values. TOPSIS is used to calculate the relative closeness to the ideal solution. Coordinated development degree model is applied to express the relationship between the relative closeness and inhomogeneity of the emergency plan, but their potential in emergency plan assessment has rarely been investigated in the literature.

The assessment results of two kinds of emergency plan by different methods such as TOPSIS and the integrated emergency plan are presented in Table 7. We find out the differences between the two methods in the emergency plans and the results are shown in Table 8. EP 2 and EP 4 gain the same ranking in both assessment methods. At the same time, the rankings of EP 1 and EP 3 are reversed by two methods. The optimal plan will change when we take into account the internal inhomogeneity of the indicators in one evaluated object.

**Table 7.** Assessment results of different methods.

| Methods | Judgment Indicator | EP 1 | EP 2 | EP 3 | EP 4 |
|---|---|---|---|---|---|
| TOPSIS method | V | 0.4305 | 0.3507 | 0.3756 | 0.3551 |
| Integrated assessment method | D | 3.5896 | 3.5771 | 3.6055 | 3.5772 |

**Table 8.** Ranking differences of emergency plans between the two methods.

| Ranking | High to Low | | | | | Optimal EP |
|---|---|---|---|---|---|---|
| Emergency plan | TOPSIS method | EP 1 | EP 3 | EP 4 | EP 2 | EP 1 |
| | Integrated assessment method | EP 3 | EP 1 | EP 4 | EP 2 | EP 3 |

The major driving factors of the water transfer project contingency plan can also be determined by the integrated assessment method. The weights of the indicators determined by the subjective weighting method and the objective weighting method are quite different, and the main driving factors cannot be determined. Therefore, we determine the main drivers based on the rankings of the indicators that affect the emergency plan. As shown in Figure 5, it can be seen that the ranking of index weights determined by subjective weighting method such as indicators C21, C32 and C33 are the main influencing factors; while the index weighting order determined by objective weighting method can be seen, indicators C23, C41 and C43 are the main influencing factors. The rankings of indicators determined by objective weighting method and subjective weighting method are not the same. The reasons why the discrepancies occurred may be the incompleteness of the data and the limitations of the evaluation experts. The combined integration weight will effectively make up for these deficiencies, and improve the weight of each index to be more reasonable, in order to show the status of this indicator in the evaluation of the whole event more accurately. In the combined integration weight ranking, indicators C33, C31, C21 and C43 are in a prominent position. These rankings indicate that indicator C33 (implementation effect) plays an important and decisive role in the evaluation of emergency plans. The main factors affecting indicator C33 are water quality level after disposal, the quantity of abandoned water and damage degree caused in the water transfer project. The level of water quality after disposal indicates the degree of pollution impact, and the quantity of abandoned water and damage degree caused in the water transfer project represent the economic losses. In the realistic practice, the main considerations for decision makers when they need to determine the emergency plan are the effect of planned disposal and the economic loss that may be caused. Therefore, the combined integration weight method proposed in this paper is not only reasonable, but also relatively scientific.

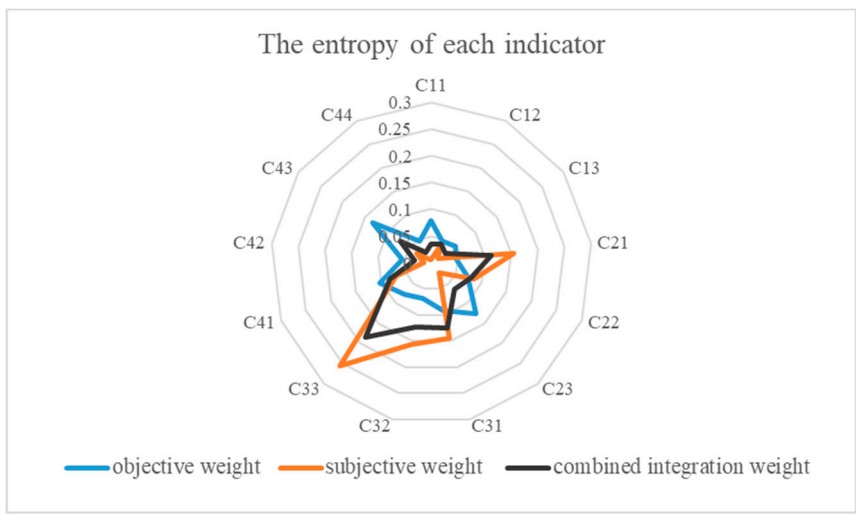

**Figure 5.** Analysis of the main driving factors of integrated emergency plan.

In the process of evaluating the four emergency plans, we consulted five relevant experts. Their evaluation results for each plan are shown in Figure 6 by different colors of the line. It can be seen from the evaluation results that the index fluctuation is small, except for individual indicators. The small fluctuations in the evaluation results of the indicators are mainly due to the fact that the evaluation experts we chosen are both familiar with environmental and water transport engineering, as well as, mainly because the knowledge level and the engineering experience between them are not too much different. Therefore, it is necessary to further improve the expert database in order to deal with other more different situations. Experts are not limited to the related scientific majors, and we should consult more staff member in many aspects, such as skilled technicians and managers in relevant projects. Comparing the four figures in Figure 6, it can be seen that the indicator C43 (contingency response to accident evolution) in the emergency plans 1 and 2 is relatively low. The main reason is that these two plans did not consider controlling pollutants in the accident pool as soon as possible, leading to an increase in the diffusion range of pollutants. Moreover, in relative terms, the variability and resilience of emergency plans 1 and 2 are both poor. That is why we did not take these two plans into the consideration of our actual project.

In the emergency plan 4, the value of indicator C23 (supplies and speed of restoring water delivery) is slightly lower. This is mainly because in emergency plan 4, in order to ensure the safety of the water transfer project, the water level before the sluice gates is required to be allowed to increase by 30 cm, resulting in a complicated control of the gates and a long time being required for the water level to stabilize. However, in actual projects, if a major pollution event occurs, the scope of pollution should be reduced at first, then secondly, the safety of the water delivery project should be considered. Therefore, comprehensive consideration shows that contingency plan 4 is not very reasonable.

In emergency plan 3, the evaluation shows that each index is relatively uniform, and there is no excessive fluctuation. There is no doubt that the experts have high recognition of emergency plan 3, so emergency plan 3 is considered to be reasonable.

In summary, although the emergency plan 1 and 2 are relatively simple to operate, (1) the gate closing time is too fast, causing great damage to the channel; (2) the pollutants are disposed in the channel, and the cost is relatively high; (3) the polluted water after the disposal treatment still below the standard level and we cannot abandon the water directly, (4) the flexibility of the program is weak. While emergency plan 3 looks rather cumbersome, the gates on the Middle Route of the South-to-North Water Transfer Project are all automated, so they're simple to operate in an actual project. The emergency plan 3 is the recommended solution in the decision support system [19]. Therefore, the integrated assessment method proposed in this paper is relatively reasonable.

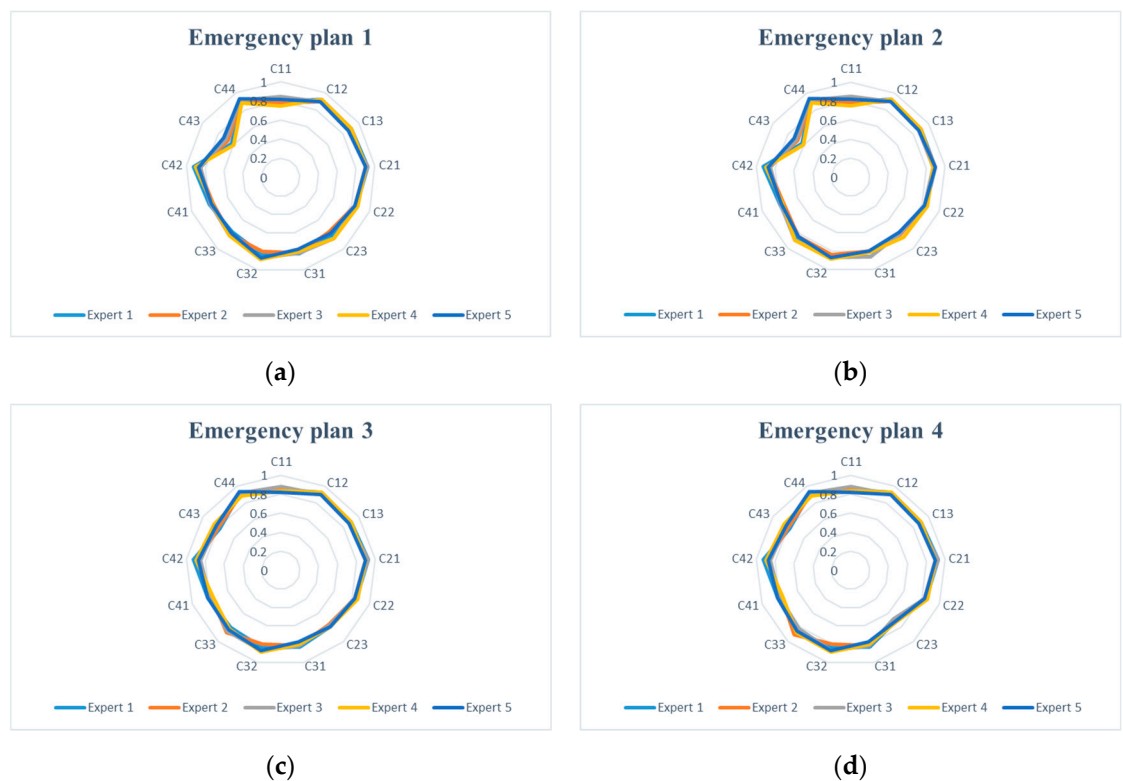

**Figure 6.** Evaluation results of various emergency plan indicators.

In this study, we introduced an improved integrated assessment method that can evaluate the emergency plan more objectively and reasonably. The main advantages of the proposed method are as follows: (1) The relative closeness to the ideal solution using TOPSIS and the inhomogeneity of all indicators based on the Shannon entropy method are synthetically considered to make the evaluation results closer to the actual situation. (2) The entropy method that considers the objective attributes to the data and the AHP method that takes into account the subjective characteristics of the data are combined, and the final determined index weight is more reasonable.

However, the proposed method also has certain deficiencies. (1) The optimal emergency plan is still limited by the experts' experience and knowledge to some extent. Therefore, we will further improve the expert database to reduce the impact of professional knowledge and experience. (2) Although the indicator system can reflect the relationship between human society and the environment, it was imperfectly scientific and reasonable. Therefore, we will explore the multi-faceted factors affecting the effectiveness of the emergency plan and establish a more perfect indicator system in further research.

## 4. Conclusions

The study proposed a novel method for assessing emergency plans based on improved TOPSIS, Shannon entropy and the Coordinated development degree model. The main conclusions are as follows:

(1) Thirteen assessment indices were selected to establish the indicator system based on the principle of being scientific, systematic, comprehensive, hierarchical, regional and dynamic. The indicator system was analyzed layer by layer, which reflects the relationship between human beings, society, the environment and transfer projects.

(2) If only the subjective method or objective method is used to determine the index weight, the result is somewhat unreasonable. This study proposes a comprehensive weighting method based on the entropy method and AHP method. The key of the comprehensive weighting method is to

combine the entropy method considering the objective attribute of the data with the AHP considering the subjective characteristics of the data, and the final determined index weight is more reasonable.

(3) An integrated assessment method was developed based on TOPSIS, Shannon entropy and the Coordinated development degree model to assess the emergency plan for sudden water pollution accidents. In order to verify the proposed method for emergency plan assessment, TOPSIS method was compared with the integrated assessment method.

(4) Using the integrated assessment method, it can be determined that the implementation effect is the main driving factor for the emergency plan optimization, which will provide decision makers with a highly scientific and reasonable plan planning and application reference.

(5) With the support of decision support system, this integrated assessment method was applied to the emergency plan optimization in Middle Route of the South-to-North Water Transfer Project. The results showed that the proposed method is feasible and provides the most reasonable results. The research results provide a new method for emergency plan assessment, which can obtain valuable information for the management, pollution control and disposal of sudden water pollution within and outside the study area.

(6) Through a series of analyses, we can conclude that the method of this study compensates for the shortcomings of the previous method. It integrates the three aspects of environment, human society and economy, and has the advantages of simplicity and thoroughness. Overall, this paper presents a successful emergency plan assessment method that can be used for emergency plan assessment of water transfer projects.

(7) Some directions for future studies are also proposed: ① The indicator system shows imperfect scientific and reasonable features, even though a large number of scientific studies have used this system as a reference and have considered the relationship between human beings, society, the environment and transfer project. Therefore, building a more comprehensive index system presents a potential avenue for future research. ② The optimal of emergency plan is still limited by the experts' experience and knowledge to some extent. Therefore, we will further improve the expert database to reduce the impact of professional knowledge and experience.

**Supplementary Materials:** The following are available online at http://www.mdpi.com/2071-1050/11/2/510/s1, Table S1: Expert score sheet for emergency plan 1; Table S2: Expert score sheet for emergency plan 2; Table S3: Expert score sheet for emergency plan 3; Table S4: Expert score sheet for emergency plan 4; Table S5: Expert score statistics table; Table S6: Calculation of the local weights of events $B_1$-$B_4$; Table S7: Calculation of the local weights and global weights of events $C_{11}$-$C_{13}$; Table S8: Calculation of the local weights and global weights of events $C_{21}$-$C_{23}$; Table S9: Calculation of the local weights and global weights of events $C_{31}$-$C_{33}$; Table S10: Calculation of the local weights and global weights of events $C_{41}$-$C_{44}$. The supplementary data to this article can be found in Appendix A.

**Author Contributions:** Y.L. (Yan Long) analyzed data formally, proposed the research, conducted fieldwork, and drafted the original manuscript. Y.Y. and Y.L. (Youming Li) assisted Y.L. (Yan Long) in data collection and research. X.L. supervised the overall research project, acquired funding to support this research. Y.T. significantly contributed to the editing of the manuscript.

**Funding:** This research was funded by [National Key R&D Program of China] grant number [2017YFC0405900], [National Science Foundation of China under Grants] grant number [51609258], and [Major National Science and Technology Project] grant number [2017ZX07108-001].

**Acknowledgments:** We thank Ivana Vuleta, editor of MDPI, for editing the English text of a draft of this manuscript.

**Conflicts of Interest:** The authors declare no conflict of interest.

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
