# Peer review of "Integrated Assessment Method of Emergency Plan for Sudden Water Pollution Accidents Based on Improved TOPSIS, Shannon Entropy and a Coordinated Development Degree Model"

_sustainability, doi:10.3390/su11020510_

Round 1
Reviewer 1 Report
General Overview
Authors of this manuscript have presented a study related to Integrated assessment method of emergency plan for sudden water pollution accidents based on improved TOPSIS, Shannon entropy. The author has proposed an integrated assessment method by incorporating improved technique for order preference by similarity to ideal solution (TOPSIS). The author has found that the method can provide targeted reference or guidance for emergency control decision makers.The topic is interesting, an important issue and generally well written, well structured and contributes to the existing knowledge. However, there are still some occasional grammar errors through the manuscript especially the article ‘’the’’, ‘’a’’ and ‘’an’’ is missing in many places, please make a spellchecking.
The results and discussion section needs further improvement, compare your findings with the other author's conclusions. Please provide more deep discussion about your results, compare your findings with the other author findings. Please clearly state the objectives and hypotheses of this study. Please check the reference style, some of the references are not according to the journal style, especially the journals abbreviations. As said, it could be a failure to comprehend the main argument; and lightening the discussion would enhance comprehensibility and make the authors’ point clearer and stronger. Therefore, the reviewer recommends to further improve the manuscript before accepting it for publication. Some of the specific comments are listed below.
Specific Comments
Avoid using the abbreviation in the abstract.
Figure 3, please improve the resolution and the text size.
The objectives of the study are not explicitly stated.
In many places’ articles ‘’the’’, ‘’a’’, ‘’an’’ is missing, please check and correct.
Please provide deeper discussing by comparing your findings with the literature.
There is a lack of references; please consider citing the following literature:
Kuriqi, A. (2014). Simulink application on dynamic modeling of biological waste water treatment for aerator tank case. International Journal Of Scientific & Technology Research, 3(11), 69-72.
Jia, R.; Jiang, X.; Shang, X.; Wei, C. Study on the Water Resource Carrying Capacity in the Middle Reaches of the Heihe River Based on Water Resource Allocation. Water 2018, 10, 1203
Kuriqi, A., Kuriqi, I., & Poci, E. (2016). Simulink Programing for Dynamic Modelling of Activated Sludge Process: Aerator and Settler Tank Case. Fresen. Environ. Bull, 25(8), 2891.
Jia, Y.; Shen, J.; Wang, H.; Dong, G.; Sun, F. Evaluation of the Spatiotemporal Variation of Sustainable Utilization of Water Resources: Case Study from Henan Province (China). Water 2018, 10, 554.
Conclusion sections is a little bit weak; please improve by adding more information related to what the authors conclude in this work and what’s the novelty or the main contribution in state of the art. This section should be more explicit and formulated. In this section, the main limitations, challenging issues, and future research orientations for similar research settings by other authors worldwide can be briefly and collectively reiterated.
Please check the references style, some of the references are not according to the journal style, especially the journals abbreviations.
Concluding Remarks
The work presented in this manuscript is an interesting topic, it needs some more efforts to improve it further. Reviewer recommend minor revision of this manuscript and publishing it only after specific improvement of the current version are made.
Author Response
Point 1: Avoid using the abbreviation in the abstract. Response: Thank you for your comment and sorry for our unclear report. We have replaced the abbreviations in the abstract and corrections have been made in manuscript. Abstract: In recent years, sudden water pollution accidents have occurred frequently. The emergency response plan optimization is the key to handling accidents. In this paper, an integrated assessment method is proposed by incorporating improved technique for order preference by similarity to ideal solution, Shannon entropy and Coordinated development degree model to evaluate emergency plan. The Shannon entropy method was used to analyze different types of index values. TOPSIS is used to calculate the relative closeness to the ideal solution. Coordinated development degree model is applied to express the relationship between the relative closeness and inhomogeneity of emergency plan. This method is tested in the decision support system of Middle Route Construction and Administration Bureau, China. By considering the different nature of the indicators, the integrated assessment method is eventually proven as a highly realistic method for assessing emergency plan. The advantages of this method are more prominent when there are more indicators of the evaluation object and the nature of each indicator is quite different. In summary, this integrated assessment method can provide targeted reference or guidance for emergency control decision makers. Keywords: Coordinated development degree model; Technique for order preference by similarity to ideal solution; Shannon entropy; Sudden water pollution; Integrated assessment; Inhomogeneity Point 2: Figure 3, please improve the resolution and the text size. Response: Thank you for your comment and sorry for our unclear report. We have improved the resolution and the text size in manuscript. Figure3. Schematic of the test area. Point 3: In many places’ articles ‘’the’’, ‘’a’’, ‘’an’’ is missing, please check and correct. Response: Thank you for your rigorous remind. We have checked and revised the paper throughout, the changes have been marked in the text. Point 4: Please provide deeper discussing by comparing your findings with the literature. Response: Thank you for your comments and all your comments are very important. We have made a comprehensive revision of the discussion. The detailed description had been added in “3.3. Discussion”. 3.3. Discussion This work presents an approach for assessing the emergency plan by combining improved TOPSIS, Shannon entropy and Coordinated development degree model. The Shannon entropy method was used to analyze different types of index values. TOPSIS is used to calculate the relative closeness to the ideal solution. Coordinated development degree model is applied to express the relationship between the relative closeness and inhomogeneity of the emergency plan, but their potential in emergency plan assessment has rarely been investigated in the literature. The assessment results of two kinds of emergency plan by different methods such as TOPSIS and the integrated emergency plan are presented in Table 7. We find out the differences between the two methods in the emergency plans and the results are shown in Table 8. EP 2 and EP 4 gain the same ranking in both assessment methods. While, the rankings of EP 1 and EP 3 are reversed by two methods. The optimal plan will change when we take into account the internal inhomogeneity of the indicators in one evaluated object. Table 7. Assessment results of different methods. Methods Judgment indicator EP 1 EP 2 EP 3 EP 4 TOPSIS method V 0.4305 0.3507 0.3756 0.3551 Integrated assessment method D 3.5896 3.5771 3.6055 3.5772 Table 8. Ranking differences of emergency plans between the two methods Ranking High to low Optimal EP Emergency plan TOPSIS method EP 1 EP 3 EP 4 EP 2 EP 1 Integrated assessment method EP 3 EP 1 EP 4 EP 2 EP 3 The major driving factors of the water transfer project contingency plan can also be determined by the integrated assessment method. The weights of the indicators determined by the subjective weighting method and the objective weighting method are quite different, and the main driving factors cannot be determined. Therefore, we determine the main drivers based on the rankings of the indicators that affect the emergency plan. As shown in Figure 5, it can be seen that the ranking of index weights determined by subjective weighting method such as indicators C21, C32 and C33 are the main influencing factors; while the index weighting order determined by objective weighting method can be seen, indicators C23, C41 and C43 are the main influencing factors. The rankings of indicators determined by objective weighting method and subjective weighting method are not the same. The reasons why the discrepancies occurred may be the incompleteness of the data and the limitations of the evaluation experts. The combined integration weight will effectively make up for these deficiencies, and improve the weight of each index more reasonable, in order to show the status of this indicator in the evaluation of the whole event more accurately. In the combined integration weight ranking, indicators C33, C31, C21 and C43 are in a prominent position. These rankings indicate that indicator C33 (implementation effect) plays an important decisive role in the evaluation of emergency plans. The main factors affecting indicator C33 are water quality level after disposal, the quantity of abandoned water and damage degree caused in the water transfer project. The level of water quality after disposal indicates the degree of pollution impact, and the quantity of abandoned water and damage degree caused in the water transfer project represent the economic losses. In the realistic practice, the main considerations for decision makers when they need to determine the emergency plan are the effect of planned disposal and the economic loss may cause. Therefore, the combined integration weight method proposed in this paper is not only reasonable, but also relatively scientific. Figure 5. Analysis of the main driving factors of integrated emergency plan. In the process of evaluating the four emergency plans, we consulted five relevant experts. Their evaluation results for each plan are shown in Figure 6 by different colors of the line. It can be seen from the evaluation results that the index fluctuation is small, except for individual indicators. The small fluctuations in the evaluation results of the indicators are mainly due to the fact that the evaluation experts we chosen are both familiar with environmental and water transport engineering, as well as, mainly because the knowledge level and the engineering experience between them are not too much different. Therefore, it is necessary to further improve the expert database in order to deal with other more different situations. Experts are not limited to the related majors in scientific, and we should consult more staff member in many aspects, such as skilled technicians and managers in relevant projects. Comparing the four figures in Figure 6, it can be seen that the indicator C43 (contingency response to accident evolution) in the emergency plans 1 and 2 is relatively low. The main reason is that these two plans did not consider controlling pollutants in the accident pool as soon as possible, leading to increase the diffusion range of pollutants. Moreover, relatively, the variability and resilience of emergency plans 1 and 2 are both poor. That is why we did not take these two plans into the consideration of our actual project. In the emergency plan 4, the value of indicator C23 (supplies and speed of restoring water delivery) is slightly lower. Mainly because in the emergency plan 4, in order to ensure the safety of the water transfer project, the water level before the sluice gates is required to be allowed to increase 30 cm, resulting in complicated control of the gates and a long time required for the water level to stabilize. However, in actual projects, if a major pollution event occurs, the scope of pollution should be reduced at first, then secondly, the safety of the water delivery project should be considered. Therefore, comprehensive consideration is that the contingency plan 4 is not very reasonable. In the emergency plan 3, it shows that the evaluation of each index is relatively uniform, and there is no excessive fluctuation. There is no doubt that the experts have high recognition of the emergency plan 3, so the emergency plan 3 is considered reasonable. In summary, although the emergency plan 1 and 2 are relatively simple to operate, (1) the gate closing time is too fast, causing great damage to the channel; (2) the pollutants are disposed in the channel, and the cost is relatively high; (3) the polluted water after the disposal treatment still below the standard, we cannot abandon the water directly, (4) the flexibility of the program is weak. While the emergency plan 3 looks rather cumbersome, but the gates on the Middle Route of the South-to-North Water Transfer Project are all automated, so it’s simple to operate in actual project. The emergency plan 3 is the recommended solution in the decision support system [19]. Therefore, the integrated assessment method proposed in this paper is relatively reasonable. (a) (b) (c) (d) Figure 6. Evaluation results of various emergency plan indicators. In this study, we introduced the improved integrated assessment method that can evaluate the emergency plan more objectively and reasonably. The main advantages of the proposed method are as follows: (1) The relative closeness to the ideal solution using TOPSIS and the inhomogeneity of all indicators based on the Shannon entropy method are synthetically considered to make the evaluation results closer to the actual situation. (2) The entropy method that considers the objective attributes to the data and the AHP method that takes into account the subjective characteristics of the data are combined, and the final determined index weight is more reasonable. However, the proposed method also has certain deficiencies. (1) The optimal of emergency plan is still limited by the experts' experience and knowledge to some extent. Therefore, we will further improve the expert database to reduce the impact of professional knowledge and experience. (2) Although the indicator system can reflect the relationship between human society and the environment, it was imperfect scientific and reasonable. Therefore, we will explore the multi-faceted factors affecting the effectiveness of the emergency plan and establish a more perfect indicator system in further research. Point 5: There is a lack of references; please consider citing the following literature: Response: Thanks for the reviewer’s suggestion. We also cited two literature mentioned by the reviewer. The detailed description had been added in “1. Introduction”. The newly added references are as follows: [4] Jia, R., Jiang, X., Shang, X., & Wei, C. Study on the Water Resource Carrying Capacity in the Middle Reaches of the Heihe River Based on Water Resource Allocation. Water, 2018, 10, 1203. [9] Jia, Y., Shen, J., Wang, H., Dong, G., & Sun, F. Evaluation of the Spatiotemporal Variation of Sustainable Utilization of Water Resources: Case Study from Henan Province (China). Water, 2018, 10, 554. Point 6: Conclusion sections is a little bit weak; please improve by adding more information related to what the authors conclude in this work and what’s the novelty or the main contribution in state of the art. This section should be more explicit and formulated. In this section, the main limitations, challenging issues, and future research orientations for similar research settings by other authors worldwide can be briefly and collectively reiterated. Response: Thank you for your comments and all your comments are very important. We have made a comprehensive revision of the conclusion. The detailed description had been added in “4. Conclusions”. 4. Conclusions The study proposed a novel method for assessing emergency plan based on improved TOPSIS, Shannon entropy and Coordinated development degree model. The main conclusions are as follows: (1) Thirteen assessment indices were selected to establish the indicator system based on the principle of being scientific, systematic, comprehensive, hierarchical, regional and dynamic. The indicator system was analyzed layer by layer, which reflect the relationship between human beings, society, the environment and transfer project. (2) If only the subjective method or objective method is used to determine the index weight, the result is somewhat unreasonable. This study proposes a comprehensive weighting method based on entropy method and AHP method. The key of the comprehensive weighting method is to combine the entropy method considering the objective attribute of the data with the AHP considering the subjective characteristics of the data, and the final determined index weight is more reasonable. (3) An integrated assessment method was developed based on TOPSIS, Shannon entropy and Coordinated development degree model to assess the emergency plan for sudden water pollution accidents. In order to verify the proposed method for emergency plan assessment, TOPSIS method was compared with the integrated assessment method. (4) Using the integrated assessment method, it can be determined that the implementation effect is the main driving factor for the emergency plan optimization, which will give decision makers with a highly scientific and reasonable plan planning and application reference. (5) With the support of decision support system, this integrated assessment method was applied to the emergency plan optimization in Middle Route of the South-to-North Water Transfer Project. The results showed that the proposed method is feasible and comes out the most reasonable results. The research results provide a new method for emergency plan assessment, which can provide valuable information for the management, pollution control and disposal of sudden water pollution within and outside the study area. (6) Through a series of analyses, we can conclude that the method of this study compensates for the shortcomings of the previous method. It integrates the three aspects of environment, human society and economy, and has the advantages of simplicity and thoroughness. Overall, this paper presents a successful emergency plan assessment method that can be used for emergency plan assessment of water transfer projects. (7) Some direction for future studies are also proposed: ① The indicator system shows imperfect scientific and reasonable even though a large number of scientific studies have used this system as a reference and have considered the relationship between human beings, society, the environment and transfer project. Therefore, building a more comprehensive index system presents a potential avenue for future research. ② The optimal of emergency plan is still limited by the experts' experience and knowledge to some extent. Therefore, we will further improve the expert database to reduce the impact of professional knowledge and experience. Point 7: Please check the references style, some of the references are not according to the journal style, especially the journals abbreviations. Response: Thanks for the reviewer’s suggestion. We have checked all references and revised them according to the journal's requirements, especially the journal's abbreviation. The detailed description had been added in “References”. Special thanks to you for your good comments.
Reviewer 2 Report
The paper's approach and conclusions appear to be insufficiently consolidated, leaving doubts to the reader about the practical value of the work. Therefore, the interest of the methodology mentioned in the paper, compared with others, should be better sustained.
The remission to "electronic supplementary material" of part of the evaluation in the case study do not seem very suitable in the final published paper. The authors are encouraged to ponder this question. Some calculations or results may possibly be partially shown in the text as examples and others not presented in the paper.
It was interesting to include some references to the type of pollutants that can be considered and their origin.
Author Response
Point 1: The paper's approach and conclusions appear to be insufficiently consolidated, leaving doubts to the reader about the practical value of the work. Therefore, the interest of the methodology mentioned in the paper, compared with others, should be better sustained.
Response: Thank you for your comments and all your comments are very important. We have made a comprehensive revision of the discussion and conclusion. The detailed description had been added in “3.3. Discussion” and “4. Conclusion”.
3.3. Discussion
This work presents an approach for assessing the emergency plan by combining improved TOPSIS, Shannon entropy and Coordinated development degree model. The Shannon entropy method was used to analyze different types of index values. TOPSIS is used to calculate the relative closeness to the ideal solution. Coordinated development degree model is applied to express the relationship between the relative closeness and inhomogeneity of the emergency plan, but their potential in emergency plan assessment has rarely been investigated in the literature.
The assessment results of two kinds of emergency plan by different methods such as TOPSIS and the integrated emergency plan are presented in Table 7. We find out the differences between the two methods in the emergency plans and the results are shown in Table 8. EP 2 and EP 4 gain the same ranking in both assessment methods. While, the rankings of EP 1 and EP 3 are reversed by two methods. The optimal plan will change when we take into account the internal inhomogeneity of the indicators in one evaluated object.
The major driving factors of the water transfer project contingency plan can also be determined by the integrated assessment method. The weights of the indicators determined by the subjective weighting method and the objective weighting method are quite different, and the main driving factors cannot be determined. Therefore, we determine the main drivers based on the rankings of the indicators that affect the emergency plan. As shown in Figure 5, it can be seen that the ranking of index weights determined by subjective weighting method such as indicators C21, C32 and C33 are the main influencing factors; while the index weighting order determined by objective weighting method can be seen, indicators C23, C41 and C43 are the main influencing factors. The rankings of indicators determined by objective weighting method and subjective weighting method are not the same. The reasons why the discrepancies occurred may be the incompleteness of the data and the limitations of the evaluation experts. The combined integration weight will effectively make up for these deficiencies, and improve the weight of each index more reasonable, in order to show the status of this indicator in the evaluation of the whole event more accurately. In the combined integration weight ranking, indicators C33, C31, C21 and C43 are in a prominent position. These rankings indicate that indicator C33 (implementation effect) plays an important decisive role in the evaluation of emergency plans. The main factors affecting indicator C33 are water quality level after disposal, the quantity of abandoned water and damage degree caused in the water transfer project. The level of water quality after disposal indicates the degree of pollution impact, and the quantity of abandoned water and damage degree caused in the water transfer project represent the economic losses. In the realistic practice, the main considerations for decision makers when they need to determine the emergency plan are the effect of planned disposal and the economic loss may cause. Therefore, the combined integration weight method proposed in this paper is not only reasonable, but also relatively scientific.
In the process of evaluating the four emergency plans, we consulted five relevant experts. Their evaluation results for each plan are shown in Figure 6 by different colors of the line. It can be seen from the evaluation results that the index fluctuation is small, except for individual indicators. The small fluctuations in the evaluation results of the indicators are mainly due to the fact that the evaluation experts we chosen are both familiar with environmental and water transport engineering, as well as, mainly because the knowledge level and the engineering experience between them are not too much different. Therefore, it is necessary to further improve the expert database in order to deal with other more different situations. Experts are not limited to the related majors in scientific, and we should consult more staff member in many aspects, such as skilled technicians and managers in relevant projects. Comparing the four figures in Figure 6, it can be seen that the indicator C43 (contingency response to accident evolution) in the emergency plans 1 and 2 is relatively low. The main reason is that these two plans did not consider controlling pollutants in the accident pool as soon as possible, leading to increase the diffusion range of pollutants. Moreover, relatively, the variability and resilience of emergency plans 1 and 2 are both poor. That is why we did not take these two plans into the consideration of our actual project.
In the emergency plan 4, the value of indicator C23 (supplies and speed of restoring water delivery) is slightly lower. Mainly because in the emergency plan 4, in order to ensure the safety of the water transfer project, the water level before the sluice gates is required to be allowed to increase 30 cm, resulting in complicated control of the gates and a long time required for the water level to stabilize. However, in actual projects, if a major pollution event occurs, the scope of pollution should be reduced at first, then secondly, the safety of the water delivery project should be considered. Therefore, comprehensive consideration is that the contingency plan 4 is not very reasonable.
In the emergency plan 3, it shows that the evaluation of each index is relatively uniform, and there is no excessive fluctuation. There is no doubt that the experts have high recognition of the emergency plan 3, so the emergency plan 3 is considered reasonable.
In summary, although the emergency plan 1 and 2 are relatively simple to operate, (1) the gate closing time is too fast, causing great damage to the channel; (2) the pollutants are disposed in the channel, and the cost is relatively high; (3) the polluted water after the disposal treatment still below the standard, we cannot abandon the water directly, (4) the flexibility of the program is weak. While the emergency plan 3 looks rather cumbersome, but the gates on the Middle Route of the South-to-North Water Transfer Project are all automated, so it’s simple to operate in actual project. The emergency plan 3 is the recommended solution in the decision support system [19]. Therefore, the integrated assessment method proposed in this paper is relatively reasonable.
In this study, we introduced the improved integrated assessment method that can evaluate the emergency plan more objectively and reasonably. The main advantages of the proposed method are as follows: (1) The relative closeness to the ideal solution using TOPSIS and the inhomogeneity of all indicators based on the Shannon entropy method are synthetically considered to make the evaluation results closer to the actual situation. (2) The entropy method that considers the objective attributes to the data and the AHP method that takes into account the subjective characteristics of the data are combined, and the final determined index weight is more reasonable.
However, the proposed method also has certain deficiencies. (1) The optimal of emergency plan is still limited by the experts' experience and knowledge to some extent. Therefore, we will further improve the expert database to reduce the impact of professional knowledge and experience. (2) Although the indicator system can reflect the relationship between human society and the environment, it was imperfect scientific and reasonable. Therefore, we will explore the multi-faceted factors affecting the effectiveness of the emergency plan and establish a more perfect indicator system in further research.
4. Conclusions
The study proposed a novel method for assessing emergency plan based on improved TOPSIS, Shannon entropy and Coordinated development degree model. The main conclusions are as follows:
(1) Thirteen assessment indices were selected to establish the indicator system based on the principle of being scientific, systematic, comprehensive, hierarchical, regional and dynamic. The indicator system was analyzed layer by layer, which reflect the relationship between human beings, society, the environment and transfer project.
(2) If only the subjective method or objective method is used to determine the index weight, the result is somewhat unreasonable. This study proposes a comprehensive weighting method based on entropy method and AHP method. The key of the comprehensive weighting method is to combine the entropy method considering the objective attribute of the data with the AHP considering the subjective characteristics of the data, and the final determined index weight is more reasonable.
(3) An integrated assessment method was developed based on TOPSIS, Shannon entropy and Coordinated development degree model to assess the emergency plan for sudden water pollution accidents. In order to verify the proposed method for emergency plan assessment, TOPSIS method was compared with the integrated assessment method.
(4) Using the integrated assessment method, it can be determined that the implementation effect is the main driving factor for the emergency plan optimization, which will give decision makers with a highly scientific and reasonable plan planning and application reference.
(5) With the support of decision support system, this integrated assessment method was applied to the emergency plan optimization in Middle Route of the South-to-North Water Transfer Project. The results showed that the proposed method is feasible and comes out the most reasonable results. The research results provide a new method for emergency plan assessment, which can provide valuable information for the management, pollution control and disposal of sudden water pollution within and outside the study area.
(6) Through a series of analyses, we can conclude that the method of this study compensates for the shortcomings of the previous method. It integrates the three aspects of environment, human society and economy, and has the advantages of simplicity and thoroughness. Overall, this paper presents a successful emergency plan assessment method that can be used for emergency plan assessment of water transfer projects.
(7) Some direction for future studies are also proposed: ① The indicator system shows imperfect scientific and reasonable even though a large number of scientific studies have used this system as a reference and have considered the relationship between human beings, society, the environment and transfer project. Therefore, building a more comprehensive index system presents a potential avenue for future research. ② The optimal of emergency plan is still limited by the experts' experience and knowledge to some extent. Therefore, we will further improve the expert database to reduce the impact of professional knowledge and experience.
Point 2: The remission to "electronic supplementary material" of part of the evaluation in the case study do not seem very suitable in the final published paper. The authors are encouraged to ponder this question. Some calculations or results may possibly be partially shown in the text as examples and others not presented in the paper.
Response: Thanks for the reviewer’s suggestion. Due to article space limitations, it is not possible to put all the supplementary materials in the manuscript, but in order to make it easier for the reader to understand our results, the more important tables have been placed in the manuscript. The detailed description had been added in “3.2. Integrated assessment method of emergency plan”.
Point 3: It was interesting to include some references to the type of pollutants that can be considered and their origin.
Response: We are very sorry for our negligence. In fact, in the indicator system proposed in the article, the indicator C21 (accuracy of potential risk assessment) already contains the type and origin of pollutants. We have already described it in detail in “2.1. Establishment of indicator system”.
2.1. Establishment of indicator system
Based on the literature review for emergency plan indictors [37-38] and the concept of decision making, which is one of the most important part of modern decision science, the indicator system for emergency plan assessment is established. The hierarchy structure of the emergency plan assessment is shown in Figure 2, which includes indicators in resistance risk (B1), timeliness (B2), economy (B3) and feasibility (B4). Resistance risk mainly refers to the ability to respond to emergencies, and resistance risk indices here are focused on comprehensiveness of prevention measures (C11), accuracy of potential risk assessment (C12) and the operator’s emergency response to danger (C13). Timeliness mainly refers to the speed of response in response to an emergency, and here are focused on speed of starting the emergency (C21), speed of arrival of personnel (C22) and supplies and speed of restoring water delivery (C23). Economy refers to the implementation effect and resource consumption. The main concern is emergency funds (C31), utilization of emergency resource (C32) and implementation effect (C33). Feasibility mainly refers to the difficulty degree of implementing emergency plan, and this paper is mainly to consider rationality and scientific of plan (C41), controllability of resource allocation (C42), contingency response to accident evolution (C43) and coordination between departments (C44). The factors affecting each indicator are shown in Table 1. When assessing the risk of pollution accidents, we need to consider the types of pollutants, origin of pollution, levels of pollutants, extent of pollution, location of accidents, economic losses, environmental impacts, and engineering damage.

Reviewer 3 Report
Title of the article
Can the title be shortened?
Abstract
In abstract, the authors have not written what the purpose of the article is, what problem they intend to solve. It would be good to write how the subject of the article is connected with the problems of sustainable development.
Introduction
The introduction is combined with a review of literature - this is not a good idea. The effect is that the problem is poorly described, research gaps are not identified (which the authors want to bring to theory), the purpose of the article is unclear (maybe it is worth giving the main goal and specific objectives?), And the literature review is superficial and nothing adds to considerations. This part of the article (theoretical part) should be ordered and extended to an orderly, logical review of literature.
The section of the article is missing, in which the authors should describe the statistical methods used and the reasons for their choice.
2.1 Establishment of indicator system
In my opinion, among the indicators for crisis situations should be a group of logistic indicators, which include the availability of material resources, human resources, information and timeliness, speed of starting the emergency, supply and speed of restoring water delivery. In crisis situations, logistics is a key factor in the effectiveness and efficiency of operations.
3.2 Integrated assessment method of emergency plan
I suggest to draw an algorithm - it will facilitate reading and understanding the description of steps.
Conclusions
Applications are worth expanding. The results of the tests are too laconic. The authors should also indicate the directions of further research, which may be based on the results obtained by them.
Bibliography
In fact, there is no world literature - the vast majority of the authors of the cited works are Asian authors, especially Chinese. It is worth pointing to the European and American achievements related to the topic of the article.
Author Response
Point 1: Title of the article Can the title be shortened?
Response: Thanks for the reviewer’s suggestion. The title of the article is that we have made discretionary determinations, which is already the most reasonable topic we think. If we shorten it further, we can't express our meaning better. So, we will continue to keep the current title.
Point 2: In abstract, the authors have not written what the purpose of the article is, what problem they intend to solve. It would be good to write how the subject of the article is connected with the problems of sustainable development.
Response: Thank you for your comment and sorry for our unclear report. We have already described it in detail in “Abstract”.
Abstract: Water is the source of all things, so it can be said that without the sustainable development of water resources, there is no sustainable development of human beings. In recent years, sudden water pollution accidents have occurred frequently. The emergency response plan optimization is the key to handling accidents. Nevertheless, the non-linear relationship between various indicators and emergency plans has greatly prevented researchers from making reasonable assessments. Thus, an integrated assessment method is proposed by incorporating improved technique for order preference by similarity to ideal solution, Shannon entropy and Coordinated development degree model to evaluate emergency plan. The Shannon entropy method was used to analyze different types of index values. TOPSIS is used to calculate the relative closeness to the ideal solution. Coordinated development degree model is applied to express the relationship between the relative closeness and inhomogeneity of emergency plan. This method is tested in the decision support system of Middle Route Construction and Administration Bureau, China. By considering the different nature of the indicators, the integrated assessment method is eventually proven as a highly realistic method for assessing emergency plan. The advantages of this method are more prominent when there are more indicators of the evaluation object and the nature of each indicator is quite different. In summary, this integrated assessment method can provide targeted reference or guidance for emergency control decision makers.
Point 3: The introduction is combined with a review of literature - this is not a good idea. The effect is that the problem is poorly described, research gaps are not identified (which the authors want to bring to theory), the purpose of the article is unclear (maybe it is worth giving the main goal and specific objectives?), And the literature review is superficial and nothing adds to considerations. This part of the article (theoretical part) should be ordered and extended to an orderly, logical review of literature.
Response: Thanks for the reviewer’s suggestion. We have revised the introduction of the manuscript based on the comments of the reviewer. The specific changes are shown in “1. Introduction”.
Point 4: The section of the article is missing, in which the authors should describe the statistical methods used and the reasons for their choice.
Response: Thank you for your comment and sorry for our negligence. We have added the reason for choosing the TOPSIS method, which is described in detail in “2.3. Calculate the relative closeness to the ideal solution based on TOPSIS”.
2.3. Calculate the relative closeness to the ideal solution based on TOPSIS
Compared with other multi-criteria decision analysis methods, TOPSIS has some advantages, such as the most adequate use of raw data information and no strict restrictions on the number of indicators [31]. TOPSIS is a ranking method that attempts to choose alternatives that simultaneously have the shortest distance from the positive ideal solution and the farthest distance from the negative ideal solution [51]. TOPSIS makes full use of the attribute information and provides a cardinal ranking of the alternatives [52]. The TOPSIS method was first proposed by C. L. Hwang and K. Yoon in 1981. The TOPSIS method is based on the closeness of a limited number of evaluation objects to the ideal solution [52]. There are two ideal solutions, one is the positive ideal solution, and the other is the negative ideal solution. The best object should be the closest to the positive ideal solution and the farthest from the negative ideal solution [51]. In this study, the relative closeness to the ideal solution is calculated based on TOPSIS.
Point 5: 2.1 Establishment of indicator system. In my opinion, among the indicators for crisis situations should be a group of logistic indicators, which include the availability of material resources, human resources, information and timeliness, speed of starting the emergency, supply and speed of restoring water delivery. In crisis situations, logistics is a key factor in the effectiveness and efficiency of operations.
Response: Thank you for your comment. We strongly agree with the reviewers' comment. However, in the establishment of the indicator system in this paper, all indicators are determined based on the emergency plan. The resistance risk, timeliness, economy and feasibility of the emergency plan are considered, separately. So, we think that both views are correct. Of course, we will conduct further research based on the comment of the reviewers. Thanks again to the reviewers for making such a great opinion.
Point 6: 3.2 Integrated assessment method of emergency plan. I suggest to draw an algorithm - it will facilitate reading and understanding the description of steps.
Response: Thank you for your comment. We have added the calculation process and algorithm in “3.2 Integrated assessment method of emergency plan.”.
3.2. Integrated assessment method of emergency plan
According to the proposed integrated assessment method of emergency plan, the above four kinds of emergency plans are evaluated, and the optimal plan is selected. The calculation process and algorithm. are shown in Figure 4 and the methodology steps together with the results are explained as follows.
Figure4. Calculation process and algorithm.
Point 7: Conclusions.--Applications are worth expanding. The results of the tests are too laconic. The authors should also indicate the directions of further research, which may be based on the results obtained by them.
Response: Thanks for the reviewer’s suggestion. We have made a comprehensive revision of the conclusion. The detailed description had been added in “4. Conclusions”.
4. Conclusions
The study proposed a novel method for assessing emergency plan based on improved TOPSIS, Shannon entropy and Coordinated development degree model. The main conclusions are as follows:
(1) Thirteen assessment indices were selected to establish the indicator system based on the principle of being scientific, systematic, comprehensive, hierarchical, regional and dynamic. The indicator system was analyzed layer by layer, which reflect the relationship between human beings, society, the environment and transfer project.
(2) If only the subjective method or objective method is used to determine the index weight, the result is somewhat unreasonable. This study proposes a comprehensive weighting method based on entropy method and AHP method. The key of the comprehensive weighting method is to combine the entropy method considering the objective attribute of the data with the AHP considering the subjective characteristics of the data, and the final determined index weight is more reasonable.
(3) An integrated assessment method was developed based on TOPSIS, Shannon entropy and Coordinated development degree model to assess the emergency plan for sudden water pollution accidents. In order to verify the proposed method for emergency plan assessment, TOPSIS method was compared with the integrated assessment method.
(4) Using the integrated assessment method, it can be determined that the implementation effect is the main driving factor for the emergency plan optimization, which will give decision makers with a highly scientific and reasonable plan planning and application reference.
(5) With the support of decision support system, this integrated assessment method was applied to the emergency plan optimization in Middle Route of the South-to-North Water Transfer Project. The results showed that the proposed method is feasible and comes out the most reasonable results. The research results provide a new method for emergency plan assessment, which can provide valuable information for the management, pollution control and disposal of sudden water pollution within and outside the study area.
(6) Through a series of analyses, we can conclude that the method of this study compensates for the shortcomings of the previous method. It integrates the three aspects of environment, human society and economy, and has the advantages of simplicity and thoroughness. Overall, this paper presents a successful emergency plan assessment method that can be used for emergency plan assessment of water transfer projects.
(7) Some direction for future studies are also proposed: ① The indicator system shows imperfect scientific and reasonable even though a large number of scientific studies have used this system as a reference and have considered the relationship between human beings, society, the environment and transfer project. Therefore, building a more comprehensive index system presents a potential avenue for future research. ② The optimal of emergency plan is still limited by the experts' experience and knowledge to some extent. Therefore, we will further improve the expert database to reduce the impact of professional knowledge and experience.
Point 8: Bibliography. In fact, there is no world literature - the vast majority of the authors of the cited works are Asian authors, especially Chinese. It is worth pointing to the European and American achievements related to the topic of the article.
Response: Thanks for the reviewer’s suggestion. We have revised the introduction and added some related achievements in European and American. The specific changes are shown in “1. Introduction”.

Round 2
Reviewer 2 Report
The authors improved the article, giving answers that can be considered minimally satisfactory to the questions posed in the previous revision.
Reviewer 3 Report
The authors have studied the comments and suggestions and have improved their article by including them. Now the article has a better version. The research methodology is well described so that the analysis is more understandable. Applications are better prepared. I therefore think that the article can be published.